# Neural Algorithmic Reasoning for Hypergraphs with Looped Transformers

## Abstract

Looped Transformers have shown exceptional neural algorithmic reasoning capability in simulating traditional graph algorithms, but their application to more complex structures like hypergraphs remains underexplored. Hypergraphs generalize graphs by modeling higher-order relationships among multiple entities, enabling richer representations but introducing significant computational challenges. In this work, we extend the Loop Transformer architecture's neural algorithmic reasoning capability to simulate hypergraph algorithms, addressing the gap between neural networks and combinatorial optimization over hypergraphs. Specifically, we propose a novel degradation mechanism for reducing hypergraphs to graph representations, enabling the simulation of graph-based algorithms, such as Dijkstra's shortest path. Furthermore, we introduce a hyperedge-aware encoding scheme to simulate hypergraph-specific algorithms, exemplified by Helly's algorithm. We establish theoretical guarantees for these simulations, demonstrating the feasibility of processing high-dimensional and combinatorial data using Loop Transformers. This work highlights the potential of Transformers as general-purpose algorithmic solvers for structured data.

## 1 Introduction

Algorithms underlie many of the technological innovations that shape our daily lives, from route planners (Hart et al., 1968) to search engines (Brin & Page, 1998). In parallel, the advent of Large Language Models (LLMs), which are built on the Transformer architecture (Vaswani et al., 2017), has dramatically extended the boundary of what is achievable through machine learning. Noteworthy systems such as GPT-4o (OpenAI, 2024b), Claude (Anthropic, 2024), and the latest OpenAI models (OpenAI, 2024c) have already propelled advances across a variety of domains, including AI agents (Chen et al., 2024d), AI search (OpenAI, 2024c), and conversational AI (Mahmood et al., 2023; Zhang et al., 2023a; Liu et al., 2024). As LLMs become increasingly capable, they open up new possibilities for integrating algorithmic reasoning into neural models.

Building on this idea, neural algorithmic reasoning (Veličković & Blundell, 2021) has recently emerged as a means to fuse the abstraction and guarantees of algorithms with the representational flexibility of neural networks. By learning to mimic the step-by-step operations of well-known procedures, neural networks can capture the generalization capabilities of algorithms. Following a self-regression pattern, (Dai et al., 2018) introduces an embedding representation for iterative graph algorithms, facilitating scalable learning of steady-state solutions. Meanwhile, (Khalil et al., 2017) employs reinforcement learning and graph embedding to automate heuristic design for NP-hard combinatorial optimization problems. By supervising the network on each intermediate state rather than only the end result, (Veličković et al., 2020) shows how learned subroutines can transfer across related tasks like Bellman-Ford algorithm (Bellman, 1958) and Prim's algorithm (Prim, 1957). Encouraging results have also been obtained in simulating graph algorithms using looped Transformers (de Luca & Fountoulakis, 2024), suggesting that Transformers may be capable of executing procedural knowledge at scale.

Extending this paradigm to more complex relational structures, such as hypergraphs, presents new opportunities and challenges. Hypergraphs naturally allow each hyperedge to contain multiple vertices, making them especially suitable for capturing higher-order interactions present in many real-world tasks. Recent studies (Xiao et al., 2022; Fatemi et al., 2023; Yang et al., 2024) illustrate how

hypergraph-based representations can enhance relational reasoning, manage complex data interdependencies, and support a larger subset of relational algebra operations. However, while standard graphs often rely on symmetrical adjacency matrices to encode connections, hypergraphs are typically represented by incidence matrices, posing additional challenges. For instance, Helly's algorithm (Helly, 1930), which assesses whether a hypergraph exhibits the Helly property (Bretto et al., 2002; Voloshin, 2002), requires operations over these incidence representations and has been shown to be NP-Hard in certain extensions (Dourado et al., 2012). Similarly, converting hypergraphs to graph-like structures often inflates feature dimensions linearly with the number of vertices due to matrix-storage demands (Bretto, 2013; Lee et al., 2020). These insights collectively raise a central question:

*Can looped Transformers achieve neural algorithmic reasoning on hypergraphs using $O(1)$ feature dimensions and $O(1)$ layers?*

**Our Contribution.** In this paper, we answer this question affirmatively by extending the results in (de Luca & Fountoulakis, 2024), which demonstrated the ability of Loop Transformers to simulate graph algorithms.

We propose and investigate the hypothesis that Loop Transformers, with their iterative processing and representational flexibility, are capable of simulating hypergraph algorithms. By leveraging the inherent capacity of Transformers for parallel processing and multi-head attention, we aim to demonstrate that these models can effectively encode and operate over hypergraph structures. The contributions of our work are outlined as follows:

- We design a degradation mechanism (Theorem 4.1) to simulate graph-based algorithms, such as Dijkstra, BFS, and DFS, on hypergraphs by dynamically degrading the incidence matrix to an adjacency matrix implicitly. This process can be simulated by a looped Transformer with constant layers and constant feature dimensions.

- We propose a novel encoding scheme to simulate Helly's algorithm (Theorem 4.2), which incorporates hyperedge-specific operations into the iterative processes of Transformers. This can also be simulated by a looped Transformer with constant layers and constant feature dimensions.

**Roadmap.** We introduce the works related to our paper in Section 2. In Section 3, we present the basic concepts, e.g., hypergraph and Loop Transformers. In Section 4, we detail our main results. In Section 5, we introduce the construction of some basic operations. In Section 6, we present some simulation results based on the basic operation. In Section 7, we give a conclusion of our paper.

## 2 RELATED WORK

### 2.1 NEURAL NETWORK EXECUTE ALGORITHMS

In recent years, the integration of neural networks with algorithmic execution has gained significant attention, leading to the emergence of a body of work aimed at improving the efficiency and performance of algorithmic tasks through deep learning techniques. This line of research focuses on leveraging the power of neural networks to either approximate traditional algorithms or directly replace them with learned models. (Siegelmann & Sontag, 1992) established the computational universality of finite recurrent neural networks with sigmoidal activation functions by proving their capability to simulate any Turing Machine. Later, (Pérez et al., 2021) proved that Transformer architectures achieve Turing completeness through hard attention mechanisms, which enable effective computation and access to internal dense representations. Building with the framework of looped transformer, (Giannou et al., 2023) uses looped transformers as the building block to make a programmable computer, showcasing the latent capabilities of Transformer-based neural networks. (de Luca & Fountoulakis, 2024) demonstrated that looped transformers can implement various graph algorithms, including Dijkstra's shortest path, Breadth-First Search (BFS), Depth-First Search (DFS), and Kosaraju's algorithm for strongly connected components, through their multi-head attention mechanism.

## 2.2 HYPERGRAPHS IN NEURAL NETWORKS

Hypergraphs have recently emerged as a powerful mathematical model for representing complex relationships in data, and they have found promising applications in deep learning. Several works have explored leveraging hypergraph-based representations to improve the performance of neural architectures in various domains. For instance, (Feng et al., 2019) introduced a hypergraph-based approach to enhance graph neural networks (GNNs), enabling better multi-way interaction modeling for tasks like node classification and link prediction, while subsequent works extended convolutional networks to hypergraphs (Yadati et al., 2019), incorporated hypergraph attention for improved interpretability (Bai et al., 2021), and leveraged hypergraphs in deep reinforcement learning for faster convergence (Bai et al., 2022). Hypergraphs have also been employed as a powerful framework for advanced reasoning. Recent research (Xiao et al., 2022; Fatemi et al., 2023; Yang et al., 2024) elucidates their capacity to refine relational inference, encapsulate intricate data interdependencies, and facilitate a broader spectrum of operations within relational algebra, thereby augmenting the expressiveness and computational efficacy of reasoning paradigms.

## 2.3 LOOPED TRANSFORMER

First introduced by (Dehghani et al., 2019), the Universal Transformer, a variant of the Transformer with a recurrent inductive bias, can be regarded as a foundational model for looped Transformers. Empirical evidence from (Yang et al., 2023) suggests that increasing the number of loop iterations enhances performance in various data-fitting tasks while requiring fewer parameters. Furthermore, (Gatmiry et al., 2024) establishes a theoretical guarantee that the looped Transformer can execute multi-step gradient descent, thereby ensuring convergence to algorithmic solutions. Recent research (Artur Back & Fountoulakis, 2024; Giannou et al., 2024) has deepened our understanding of how specific algorithms can be emulated and how their training dynamics facilitate convergence, particularly in the context of in-context learning. (Gatmiry et al., 2024; Chen et al., 2024b) showed that looped transformers can efficiently do in-context learning by multi-step gradient descent. (Giannou et al., 2023) uses Transformers as the building block to build a programmable computer, showcasing the latent capabilities of Transformer-based neural networks. Beyond (Giannou et al., 2023), (Liang et al., 2024c) proves that a looped 23-layer $\mathrm{ReLU} - \mathrm{MLP}$ is capable of performing the basic necessary operation of a programmable computer.

## 3 PRELIMINARIES

In Section 3.1, we introduce the fundamental notations used throughout this work. Section 3.2 outlines the concept of simulation, while Section 3.3 provides an overview of essential concepts related to hypergraphs. Lastly, Section 3.4 describes the architecture of the looped transformer.

## 3.1 NOTATION

We represent the set $\{1, 2, \ldots, n\}$ as $[n]$. For a matrix $A \in \mathbb{R}^{m \times n}$, the $i$-th row is denoted by $A_i \in \mathbb{R}^n$, and the $j$-th column is represented as $A_{*,j} \in \mathbb{R}^m$, where $i \in [m]$ and $j \in [n]$. For $A \in \mathbb{R}^{m \times n}$, the $j$-th entry of the $i$-th row $A_i \in \mathbb{R}^n$ is denoted by $A_{i,j} \in \mathbb{R}$. The identity matrix of size $d \times d$ is denoted by $I_d \in \mathbb{R}^{d \times d}$. The vector $\mathbf{0}_n$ denotes a length-$n$ vector with all entries equal to zero, while $\mathbf{1}_n$ denotes a length-$n$ vector with all entries equal to one. The matrix $\mathbf{0}_{n \times d}$ represents an $n \times d$ matrix where all entries are zero. The inner product of two vectors $a, b \in \mathbb{R}^d$ is expressed as $a^\top b$, where $a^\top b = \sum_{i=1}^d a_i b_i$.

## 3.2 SIMULATION

**Definition 3.1** (Simulation, Definition 3.1 in (de Luca & Fountoulakis, 2024)). *We define the following:*

- *Let $h_F : \mathcal{X} \to \mathcal{Y}$ be the function that we want to simulate.*

- *Let $h_T : \mathcal{X}' \to \mathcal{Y}'$ be a neural network.*

- *Let $g_e : \mathcal{X} \to \mathcal{X}'$ be an encoding function.*

- *Let $g_d : \mathcal{Y}' \to \mathcal{Y}$ be a decoding function.*

*We say that a neural network $h_T$ can simulate an algorithm step $h_F$ if for all input $x \in \mathcal{X}$, $h_F(x) = g_d(h_T(g_e(x)))$. We say that a looped transformer can simulate an algorithm if it can simulate each algorithm step.*

### 3.3 HYPERGRAPH

A weighted hypergraph is expressed as $H := (V, E, w)$, where $w$ is a function assigning a real-valued weight to each hyperedge. The total number of vertices in the hypergraph is $n_v := |V|$, and the total number of hyperedges is $n_e := |E|$. The vertices are assumed to be labeled sequentially from 1 to $n_v$, and the hyperedges are labeled from 1 to $n_e$.

We define the incident matrix of the weighted hypergraph as follows:

**Definition 3.2** (Incident matrix of hypergraph). *Let $H = (V, E, w)$ be a weighted hypergraph, where $V = \{v_1, v_2, \ldots, v_{n_v}\}$ is the set of vertices and $E = \{e_1, e_2, \ldots, e_{n_e}\}$ is the set of hyperedges, with $n_v = |V|$ and $n_e = |E|$. Each hyperedge $e_j \in E$ is associated with a weight $w(e_j) \in \mathbb{R}_+$ and consists of a subset of vertices from $V$. The incident matrix $A \in \mathbb{R}_+^{n_v \times n_e}$ of $H$ is defined such that the rows correspond to vertices and the columns correspond to hyperedges. For each vertex $v_i \in V$ and hyperedge $e_j \in E$, the entry $A_{i,j}$ is given by*

$$A_{i,j} = \begin{cases} w(e_j) & \text{if } v_i \in e_j, \\ 0 & \text{otherwise.} \end{cases}$$

In this paper, we use $X \in \mathbb{R}^{K \times d}$ to represent the input matrix, where $d$ is the feature dimension and $K = \max\{n_v, n_e\} + 1$ is the row number we need for simulation. To match the dimension of $X$ and incident matrix, we use a padded version of $A$, which is defined as its original entries of $A$ preserved in the bottom-right block:

**Definition 3.3** (padded version of incident matrix). *Let incident matrix of hypergraph $A \in \mathbb{R}_+^{n_v, n_e}$ be defined in Definition 3.2, let $K \in \mathbb{N}$ safisfies $K \geq \max\{n_v, n_e\} + 1$ is the row number of $X$, we define the padded version of incident matrix $A$ as:*

- **Part 1.** *If $n_e > n_v$, we define:*

$$\widetilde{A} := \begin{bmatrix} 0 & \mathbf{0}_{n_e}^\top \\ \mathbf{0}_{n_v} & A \\ \mathbf{0}_{K-n_v-1} & \mathbf{0}_{(K-n_v-1) \times n_e} \end{bmatrix}.$$

- **Part 2.** *If $n_e < n_v$, we define:*

$$\widetilde{A} := \begin{bmatrix} 0 & \mathbf{0}_{n_e}^\top & \mathbf{0}_{K-n_e-1}^\top \\ \mathbf{0}_{n_v} & A & \mathbf{0}_{n_v \times (K-n_e-1)} \end{bmatrix}.$$

- **Part 3.** *If $n_e = n_v$, we define:*

$$\widetilde{A} := \begin{bmatrix} 0 & \mathbf{0}_{n_e}^\top \\ \mathbf{0}_{n_v} & A \end{bmatrix}.$$

### 3.4 LOOPED TRANSFORMERS

Following the setting of (de Luca & Fountoulakis, 2024), we use standard transformer layer (Vaswani et al., 2017) with an additional attention mechanism that incorporates the incident matrix. The additional attention mechanism is defined as follows:

**Definition 3.4** (Single-head attention). *Let $W_Q, W_K \in \mathbb{R}^{d \times d_a}$ be the weight matrices of query and key, $W_V \in \mathbb{R}^{d \times d}$ be the weight matrix of value, and $\sigma$ be the hardmax[1] function. We define the*

---

[1]The hardmax, a.k.a., $\arg\max$, is defined by $[\sigma(\Phi)]_i := \sum_{k \in K} e_k / |K|$, where $e_k$ is the standard basis vector for any $k \in [K]$ and $K = \{k \mid \Phi_{ik} = \max\{\Phi_i\}\}$.

*single-head attention $\psi^{(i)}$ as*

$$\psi^{(i)}(X, \widetilde{A}) := \widetilde{A}\sigma(XW_Q^{(i)}W_K^{(i)^\top}X^\top)XW_V^{(i)}.$$

**Remark 3.5.** *In this paper, we set $d_a = 2$.*

The $\psi$ function is an essential construction in the definition of multi-head attention:

**Definition 3.6** (Multi-head attention). *Let $\psi$ be defined in Definition 3.4, let $\widetilde{A}$ be defined in Definition 3.3. We define the multi-head attention $\psi^{(i)}$ as*

$$f_{\text{attn}}(X, \widetilde{A}) := \sum_{i \in M_A} \psi^{(i)}(X, \widetilde{A}) + \sum_{i \in M_{A^\top}} \psi^{(i)}(X, \widetilde{A}^\top) + \sum_{i \in M} \psi^{(i)}(X, I_{n+1}) + X,$$

*where $M_A$, $M_{A^\top}$, $M$ are the index set of the attention incorporated the incident matrix, and the attention incorporated the transpose of the incident matrix, and attention heads for the standard attention which is defined in (Vaswani et al., 2017).*

**Remark 3.7.** *The total number of attention heads is $|M| + |M_A| + |M_{A^\top}|$.*

For the MLP (Multilayer Perceptron) layer, we have the following definition.

**Definition 3.8** (MLP layer). *Let $\phi$ be the ReLU function, let $W \in \mathbb{R}^{d \times d}$ be the weight of the MLP, where $d$ is the feature dimension of $X$, let $m$ be the number of layers. For $j = [m]$, we define the MLP layer as $f_{\text{mlp}}(X) := Z^{(m)}W^{(m)} + X$, where $Z^{(j+1)} := \phi(Z^{(j)}W^{(j)})$ and $Z^{(1)} := X$.*

**Remark 3.9.** *In the construction of this paper, we set the number of layers $m = 4$.*

Combine the definition of multi-head attention and MLP layer, and we show the definition of the transformer layer:

**Definition 3.10** (Transformer layer). *Let $\widetilde{A}$ be defined in Definition 3.3, let $f_{\text{attn}}$ be defined in Definition 3.6, let $f_{\text{mlp}}$ be defined in Definition 3.8. We define the transformer layer as*

$$f(X, \widetilde{A}) := f_{\text{mlp}}(f_{\text{attn}}(X, \widetilde{A})).$$

**Definition 3.11** (Multi layer Transformer). *Let $\widetilde{A}$ be defined in Definition 3.3, let $m = O(1)$ denote the layer of transformers. Let the transformer layer $f$ be defined in 3.10. We define the multi-layer transformer as $h_T(X, \widetilde{A}) := f_m \circ f_{m-1} \circ \cdot \circ f_1(X, \widetilde{A})$.*

In the construction of this paper, we set $X \in \mathbb{R}^{K \times d}$, where $K$ is defined in Definition 3.3, and $d$ is a constant independent of $K$. The matrix $X$ stores different variables in its columns. A variable can either be an array or a scalar. Scalars are stored in the top row of the corresponding column, leaving the remaining $K - 1$ rows as 0. Arrays are stored in the bottom $K - 1$ rows of the column, leaving the top row as 0. We use $B_{\text{global}}$ to represent a column where only the top scalar is 1, while the rest of the entries are 0. Conversely, we use $B_{\text{local}}$ to represent a column where the top scalar is 0, while the remaining entries are 1. Additionally, we use $P$ to represent the two columns of position embeddings, where the first column contains $\sin(\theta_i)$, and the second column contains $\cos(\theta_i)$ for $i \in [K - 1]$. Finally, $P_{\text{cur}}$ is used to represent the position embedding of the current vertex or hyperedge.

---

**Algorithm 1** Looped Transformer, Algorithm 1 in (Giannou et al., 2023)

---

1: **procedure** LOOPEDTRANSFORMER($X \in \mathbb{R}^{K \times d}, \widetilde{A} \in \mathbb{R}^{K \times K}, \text{termination} \in \{0, 1\}$)
2:     **while** $X[0, \text{termination}] = 0$ **do**
3:         $X \leftarrow h_T(X, \widetilde{A})$
4:     **end while**
5: **end procedure**

---

**Definition 3.12** (Positional Encoding). *Let $\delta$ be the minimum increment angle, and let $\widehat{\delta}$ be its nearest representable approximation. Define the rotation matrix $R_{\widehat{\delta}} \in \mathbb{R}^{2 \times 2}$ as*

$$R_{\widehat{\delta}} = \begin{bmatrix} \cos\widehat{\delta} & -\sin\widehat{\delta} \\ \sin\widehat{\delta} & \cos\widehat{\delta} \end{bmatrix}.$$

*Initialize the positional encoding with $p_0 = \begin{pmatrix} 0 \\ 1 \end{pmatrix} \in \mathbb{R}^2$. For each node $i \geq 1$, the positional encoding $p_i \in \mathbb{R}^2$ is defined recursively by $p_i = R_{\widehat{\delta}}^\top p_{i-1}$. The positional encoding for node $i$ is represented as the tuple $(p_i^{(1)}, p_i^{(2)})$, where $p_i^{(1)}$ and $p_i^{(2)}$ are the first and second components of the vector $p_i$, respectively.*

*Additionally, the maximum number of distinct nodes that can be uniquely encoded is bounded by*

$$N_{\max} = \left\lfloor \frac{2\pi}{\widehat{\delta}} \right\rfloor,$$

*which is determined by the precision of $\widehat{\delta}$.*

## 4 MAIN RESULTS

Section 4.1 resents a dynamic degradation mechanism to extend Dijkstra, BFS, and DFS from graphs to hypergraphs. In Section 4.2, we reformulate the Helly property test into a looped-Transformer-executable algorithm and proves its correctness for weighted hypergraphs. Moving on to Section 4.3, we show the looped Transformer can simulate deterministic hypergraph motif algorithms.

### 4.1 DEGRADATION

We observe that (de Luca & Fountoulakis, 2024) presents the running results of several algorithms on graphs, which primarily include Dijkstra's algorithm for the shortest path, Breadth-First Search (BFS), and Depth-First Search (DFS). (de Luca & Fountoulakis, 2024) provides a general representation of such algorithms in Algorithm 7 of their paper. Given that Dijkstra's algorithm for the shortest path, BFS, and DFS can be straightforwardly extended to hypergraphs (Gao et al., 2014) by identifying the shortest hyperedge between two nodes and treating it as the distance between those nodes, we propose a degradation mechanism in Algorithm 2. This mechanism allows dynamic access to the shortest hyperedge between two vertices without storing static adjacent matrix. Using this mechanism, we can extend Dijkstra's algorithm for the shortest path, BFS, and DFS from graphs to hypergraphs. We formally state the theorem regarding the degradation mechanism as follows:

**Theorem 4.1** (Degradation, informal version of Theorem E.1). *A looped transformer $h_T$ defined in Definition 3.11 exists, consisting of 10 layers, where each layer includes 3 attention heads with feature dimension of $O(1)$. This transformer can simulate the degradation operation (Algorithm 2) for hypergraphs, supporting up to $O(\widehat{\delta}^{-1})$ vertices and $O(\widehat{\delta}^{-1})$ hyperedges, where $\widehat{\delta}$ defined in Definition 3.12 is the nearest representable approximation of the minimum increment angle.*

### 4.2 HELLY

The Helly property in a hypergraph refers to a specific condition in the family of its hyperedges. A hypergraph is said to satisfy the Helly property if, for every collection of its hyperedges, whenever the intersection of every pair of hyperedges in the collection is non-empty, there exists at least one hyperedge in the collection that intersects all the others. This property is named after Helly's theorem in convex geometry, which inspired its application in combinatorial settings like hypergraphs. We have reformulated the algorithm from Algorithm 3 of (Bretto, 2013), which determines whether a hypergraph possesses the Helly property, into a form executable by the Looped Transformer, represented as our Algorithm 5. We now state the following theorem:

**Theorem 4.2** (Helly, informal version of Theorem E.2). *A looped transformer $h_T$ exists, where each layer is defined as in Definition 3.10, consisting of 11 layers, where each layer includes 3 attention heads with feature dimension of $O(1)$. This transformer can simulate the Helly algorithm (Algorithm 5) for weighted hypergraphs, handling up to $O(\widehat{\delta}^{-1})$ vertices and $O(\widehat{\delta}^{-1})$ hyperedges, where $\widehat{\delta}$ defined in Definition 3.12 is the nearest representable approximation of the minimum increment angle.*

---

**Algorithm 2** Degradation Iterate of hyperedge

---

1: $\text{idx}_{\text{hyperedge}} \leftarrow 0$
2: $\text{val}_{\text{hyperedge}} \leftarrow \mathbf{0}_{n_v} \| \mathbf{1}_{K-1-n_v}$
3: $\text{iszero}_{\text{hyperedge}} \leftarrow \mathbf{0}_{n_v} \| \mathbf{1}_{K-1-n_v}$
4: $\text{candidates}_{\text{hyperedge}} \leftarrow \Omega \cdot \mathbf{1}_{K-1}$
5: $\text{visit}_{\text{hyperedge}} \leftarrow \mathbf{0}_{n_e} \| \mathbf{1}_{K-1-n_e}$
6: $\text{termination}_{\text{hyperedge}} \leftarrow 0$         ▷ Initialization of visiting hyperedges
7: ___________________________________________________________________
8: **procedure** VISITHYPEREDGE($\widetilde{A} \in \mathbb{R}_+^{n_v \times n_e}$)
9:      **if** $\text{termination}_{\text{hyperedge}} = 1$ **then**
10:          $\cdots$         ▷ Re-initialization of visiting edge, Lemma C.1
11:          $\text{termination}_{\min} \leftarrow 0$
12:      **end if**
13:      $\text{idx}_{\text{hyperedge}} \leftarrow \text{idx}_{\text{hyperedge}} + 1$         ▷ Increment, Lemma C.2
14:      $\text{val}_{\text{hyperedge}} \leftarrow A[:, \text{idx}_{\text{hyperedge}}]$         ▷ Read column from incident matrix, Lemma C.7
15:      **for** $i = 1 \rightarrow K - 1$ **do**
16:          $\text{iszero}_{\text{hyperedge}}[i] \leftarrow (\text{val}_{\text{hyperedge}}[i] \leq 0)$         ▷ Compare, Lemma C.3
17:      **end for**
18:      **for** $i = 1 \rightarrow K - 1$ **do**
19:          **if** $\text{iszero}_{\text{hyperedge}}[i] = 1$ **then**
20:              $\text{val}_{\text{hyperedge}}[i] \leftarrow \Omega$         ▷ Selection, Lemma C.1
21:          **end if**
22:      **end for**
23:      **for** $i = 1 \rightarrow K - 1$ **do**
24:          **if** $\text{val}_{\text{hyperedge}}[i] < \text{candidates}[i]$ **then**         ▷ Compare, Lemma C.3
25:              $\text{candidates}[i] \leftarrow \text{val}_{\text{hyperedge}}[i]$         ▷ Update variables, Lemma C.1
26:          **end if**
27:      **end for**
28:      $\text{visit}_{\text{hyperedge}}[\text{idx}_{\text{hyperedge}}] \leftarrow 1$         ▷ Write scalar to column, Lemma C.5
29:      $\text{termination}_{\text{hyperedge}} \leftarrow \neg(0 \text{ in } \text{visit}_{\min})$         ▷ Trigger termination, Lemma C.6
30:      $\text{termination}_{\min} \leftarrow \text{termination}_{\min} \wedge \text{termination}_{\text{hyperedge}}$         ▷ AND, Lemma C.8
31: **end procedure**

---

### 4.3 FURTHURE DISCUSSION FOR THE POWER OF LOOPED TRANSFORMER

Our algorithm can be extended to the family of deterministic hypergraph motifs algorithms introduced by (Lee et al., 2020). By using a similar proof as in Theorem 4.2, we can simulate Hypergraph Projection as a preprocessing step to obtain a static projection graph and simulate the Exact H-motif Counting operation based on this static projection graph. Combining these observations, we conclude that our algorithm can effectively simulate deterministic hypergraph motifs algorithms. Specifically, Hypergraph Projection can be simulated using the selection operation in Lemma C.1 and the addition operation in Lemma C.10; Exact H-motif Counting can be simulated using the AND operation in Lemma C.8, the comparison operation in Lemma C.3, and the addition operation in Lemma C.10. Note that although our method can leverage a looped Transformer to simulate deterministic algorithms on hypergraphs, it cannot simulate algorithms involving stochastic processes or sampling, such as random sampling on hypergraphs (see Algorithm 3 in (Lee et al., 2020)).

Furthermore, we can dynamically compute the hypergraph projection in a looped process, as presented in Theorem 4.1. The key difference lies in the computational resources: the static projection graph requires $O(K)$ columns for storage due to its size dependence on $K$, which prevents a solution in $O(1)$ column space. However, if we employ a similar method to Algorithm 2, the storage cost reduces to $O(1)$ columns at the expense of requiring more iterations. Thus, our methods may cover all sets of deterministic hypergraph motifs and show that the looped transfer is powerful.

# 5 KEY OPERATION IMPLEMENTATION

In this section, we introduce some basic operations that can be simulated by a single-layer transformer, which serve as fundamental primitives in our theoretical framework. Using these primitives, we can simulate complex deterministic algorithms that use hypergraphs or hyperedges as iterative objects.

**Selection.** Here, we present the selection operation. Given a boolean array, referred to as condition $C$, a target array $E$, and two value arrays, $V_0$ and $V_1$, the selection operation can achieve the following: when the condition $C$ is 1, the value from $V_1$ will be written to the target array $E$, when the condition $C$ is 0, the value from $V_0$ will be written to the target array $E$. By this operation, we can combine two arrays into a single array. Furthermore, by setting the target array $E$ to be the same as one of the value arrays, i.e., $V_0$ or $V_1$, we can realize conditional updates. This means that the value of an array will be updated if and only if a certain condition holds.

If $V_0$, $V_1$, $E$, and $C$ are scalars, this operation can be directly applied to scalars, as the lower $[K-1]$ values are kept as 0. In the following lemma, we aim to show that the selection operation can be simulated by a single-layer transformer. See details in Lemma C.1.

**Increment.** Here, we present the operation of increment using positional embeddings. Let $C_1$ and $C_2$ represent the positional embeddings corresponding to $\sin(\cdot)$ and $\cos(\cdot)$, respectively. The primary objective is to employ a rotation matrix to update the angles encoded in $C_1$ and $C_2$ by increasing the angular offset $\widehat{\delta}$. The updated values are then written to the target locations $D_1$ and $D_2$. Typically, when

$$C_1 = D_1 \tag{1}$$

and

$$C_2 = D_2 \tag{2}$$

, this operation can be performed in-place. See details in Lemma C.2.

**Comparason.** Here, we introduce the operation of comparison, which involves comparing two arrays. In greedy algorithms, conditional updates are often employed, where the stored optimal solution is updated only if the current solution is better than the previously stored one. This process is captured in the condition update within the selection operation, as described in Lemma C.1. Notably, this requires a Boolean array as the condition input. For example, in Dijkstra's algorithm, we evaluate whether a path is shorter and update only those paths that are shorter than the currently stored shortest paths. See details in Lemma C.3.

**Read Scalar From Column.** We present the operation of reading a scalar from a column. Let the target column be denoted as $E$, the position embeddings of the source row $C$ as $C_1$ and $C_2$, and the source column as $D$. The array is stored in column $D$, and our objective is to extract the scalar at the $C$-th position and write it to the top row of column $E$. In an MLP layer represented as $XW$, we can extract a column using its column index. However, to extract a specific row index, we leverage the property of the positional embedding,

$$p_i^\top p_j < p_i^\top p_i \tag{3}$$

for $i \neq j$. This operation enables more flexible operations on individual values within the array. See details in Lemma C.4.

**Write Scalar to Column.** We present the operation of writing a scalar to a column. This operation parallels the process of reading a scalar from a column. The construction in this step also employs position embedding, analogous to Lemma C.4. See details in Lemma C.5.

**Termination.** In the greedy algorithm, we terminate the process and return the result after traversing all objects. Here, we maintain an array, marking the traversed objects as 1. Notably, since the value of

$$K - 1 \tag{4}$$

does not always equal the number of objects (typically $n_v$ or $n_e$), we fill the remaining positions with 1. See details in Lemma C.6.

**Read from Incident Matrix.** Instead of storing a static incidence matrix in a matrix $X \in \mathbb{R}^{K \times d}$, we utilize an attention-head-incorporated incidence matrix as defined in Definition 3.6. This construction allows $d$ to be a constant independent of $n_e$ and $n_v$, implying that the feature dimension of the model $h_T$ can be controlled in $O(1)$. See details in Lemma C.7.

**AND.** This operation can be applied to both arrays and scalars because, even when the input column is always $0$, the result of the AND operation will remain $0$. This operation is particularly useful when combining two conditions to trigger a specific operation. See details in Lemma C.8.

**Repeat AND.** Different from the regular AND operation described in Lemma C.8, the repeat AND operation combines a scalar and an array. It can be understood as first replicating the scalar into a length-$K-1$ array, followed by performing the regular AND operation. This operation is often used in nested if statements. See details in Lemma C.9.

**Repeat Addition.** Similar to the repeat AND operation, the repeat addition operation first replicates a scalar into an array and then performs element-wise addition between two arrays. This operation is commonly used when updating the distance to the next vertices. See details in Lemma C.10.

# 6 SIMULATION

We present the simulation result of visiting hyperedges iteratively in Section 6.1. We discuss the simulation result of Dijkstra's Algorithm in Section 6.2.

## 6.1 ITERATION OF VISITING HYPEREDGES

First, we present our result on visiting hyperedges iteratively. See details in Algorithm 2.

**Lemma 6.1** (Visiting hyperedges iteratively, informal version of Lemma D.2). *A looped transformer $h_T$ exists, where each layer is defined as in Definition 3.10, consisting of 10 layers where each layer includes 3 attention heads with feature dimension of $O(1)$. This transformer simulates the operation of iteratively visiting hyperedges for weighted hypergraphs, accommodating up to $O(\widehat{\delta}^{-1})$ vertices and $O(\widehat{\delta}^{-1})$ hyperedges.*

## 6.2 DIJKSTRA'S ALGORITHM

Furthermore, we combine the iteratively visiting hyperedge pattern with Dijkstra's Algorithm to extend it to a hypergraph. For details, see Algorithm 4.

**Theorem 6.2** (Dijkstra's Algorithm on hypergraph, informal version of Theorem D.1). *A looped transformer $h_T$ exists, where each layer is defined as in Definition 3.10, consisting of 27 layers, where each layer includes 3 attention heads with feature dimension of $O(1)$. This transformer simulates Dijkstra's Algorithm iteratively for weighted hypergraphs, supporting up to $O(\widehat{\delta}^{-1})$ vertices and $O(\widehat{\delta}^{-1})$ hyperedges.*

# 7 CONCLUSION

In this work, we extended the capabilities of Loop Transformers to the domain of hypergraphs, addressing the computational challenges posed by their complex structures. By introducing a degradation mechanism to simulate graph-based algorithms on hypergraphs and a hyperedge-aware encoding scheme for hypergraph-specific algorithms, we demonstrated the feasibility of using Transformers for hypergraph algorithm simulation. Our results, supported by theoretical guarantees, underscore the potential of Loop Transformers as general-purpose computational tools capable of bridging neural networks and combinatorial optimization tasks on structured, high-dimensional data. These findings not only expand the applicability of Transformers but also open new avenues for solving real-world problems.

## ETHIC STATEMENT

This paper does not involve human subjects, personally identifiable data, or sensitive applications. We do not foresee direct ethical risks. We follow the ICLR Code of Ethics and affirm that all aspects of this research comply with the principles of fairness, transparency, and integrity.

## REPRODUCIBILITY STATEMENT

We ensure reproducibility of our theoretical results by including all formal assumptions, definitions, and complete proofs in the appendix. The main text states each theorem clearly and refers to the detailed proofs. No external data or software is required.

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

# Appendix

**Roadmap.** In Section A, we introduce more related work. In Section B, we introduce some tools from previous work. In Section C, we present the missing lemma in Section 5. In Section D, we present the missing proof in Section 6. In Section E, we present the missing proof in Section 4. We discuss our impact statement in Section F.

## A    MORE RELATED WORK

### A.1    LARGE LANGUAGE MODELS

Transformer-based neural networks (Vaswani et al., 2017) have rapidly become the leading framework for natural language processing within machine learning. When these models scale to billions of parameters and are trained on expansive, heterogeneous data, they are frequently called large language models (LLMs) or foundation models (Bommasani et al., 2021). Representative LLMs include BERT (Devlin et al., 2019), PaLM (Chowdhery et al., 2022), Llama (Touvron et al., 2023), ChatGPT (OpenAI, 2024a), and GPT4 (OpenAI, 2023). Such models exhibit versatile capabilities (Bubeck et al., 2023) across a broad array of downstream tasks.

In pursuit of optimizing LLMs for specific applications, a variety of adaptation strategies have been introduced. These range from using adapters (Hu et al., 2022; Zhang et al., 2023b; Gao et al., 2023a; Shi et al., 2023), calibration methods (Zhao et al., 2021; Zhou et al., 2023), and multitask fine-tuning (Gao et al., 2021a; Xu et al., 2023; Von Oswald et al., 2023; Xu et al., 2024c), to prompt tuning (Gao et al., 2021b; Lester et al., 2021), scratchpad techniques (Nye et al., 2021), instruction tuning (Li & Liang, 2021; Chung et al., 2022; Mishra et al., 2022), symbol tuning (Wei et al., 2023), black-box tuning (Sun et al., 2022), reinforcement learning from human feedback (Ouyang et al., 2022), chain-of-thought reasoning (Wei et al., 2022; Khattab et al., 2022; Yao et al., 2023; Zheng et al., 2024), and more.

Recent relevant research includes works on tensor transformers (Sanford et al., 2024; Alman & Song, 2024a; Liang et al., 2024f; Li et al., 2024c; Zhang et al., 2025), acceleration techniques (Xu et al., 2024a; Wu et al., 2024b; Shen et al., 2024a; Liang et al., 2024a; Qin et al., 2023; Song et al., 2024; Hu et al., 2024a; Shen et al., 2024b; Shi et al., 2024; Wu et al., 2024a; Ke et al., 2025; Hu et al., 2024b;c; Li et al., 2024e; Hu et al., 2024f; Chen et al., 2024e; Ke et al., 2024; Liang et al., 2024b; Li et al., 2024d; Hu et al., 2024e; Liang et al., 2024d; Li et al., 2024f; Hu et al., 2023; Li et al., 2024b; Alman & Song, 2024b; Chen et al., 2024c), and other related studies (Demirel et al., 2022; Shrivastava et al., 2023; Gao et al., 2023c; Liang et al., 2024e; Song & Yang, 2023; Chen et al., 2024a; Xu et al., 2024b; Li et al., 2024g; Deng et al., 2022; Gao et al., 2023b; Chen et al., 2024f; Li et al., 2024a; Hu et al., 2024d; Sinha et al., 2023; Zhang et al., 2024; Zhang, 2024; Li et al., 2025).

## B    TOOLS FROM PREVIOUS WORK

Here, we present the lemma of Iteratively visiting to find minimum value.

**Lemma B.1** (Get minimum value, Implicitly in (de Luca & Fountoulakis, 2024)). *If the following conditions hold:*

- *Let the transformer be defined as Definition 3.10.*

*Then, we can show that a 7-layer transformer can simulate the operation of getting the minimum value in Algorithm 3.*

## C    MISSING PROOF IN KEY OPERATION IMPLEMENTATION

In Section C.1, the operation of selection is discussed. In Section C.2, the operation of increment is discussed. In Section C.4, the operation of reading scalar from column is discussed. In Section C.3, the operation of comparison is discussed. In Section C.5, the operation of writing scalar to column

is discussed. In Section C.6, the operation of termination is discussed. In Section C.7, the operation of reading from incident matrix is discussed. In Section C.8, the operation of AND is discussed. In Section C.9, the operation of repeat AND is discussed. In Section C.10, the operation of repeat addition is discussed.

## C.1 SELECTION

**Lemma C.1** (Selection). *If the following conditions hold:*

- *Let the transformer be defined as Definition 3.10.*

- *Let $\Omega$ represent the maximum absolute value within a clause.*

- *Let $V_0, V_1$ be the index for clause values, where $X[i, V_0], X[i, V_1] \in [-\Omega, \Omega]$ for $i \in [K]$.*

- *Let $C$ be the index for conditions, where $X[i, C] \in \{0, 1\}$ for $i \in [K]$.*

- *Let $E$ be the index of the target field.*

*Then we can show that a single-layer transformer can simulate a selection operation, which achieves the following: if $X[i, C] = 1$, the value $X[i, V_1]$ is written to $X[i, E]$; otherwise, the value $X[i, V_0]$ is written to $X[i, E]$, for all $i \in [K]$.*

*Proof.* Because only the MLP layer is needed, we set all the parameters in the attention layer to 0 while keeping the residual connection. Let $S_1$, $S_2$ be the index for the scratchpad. We construct the weights in MLP as follows:

$$(W^{(1)})_{a,b} = \begin{cases} 1 & \text{if } (a, b) \in \{(V_0, V_0), (V_1, V_1), (C, C), (E, E), (B_{\text{global}}, B_{\text{global}}), (B_{\text{local}}, B_{\text{local}})\}; \\ 0 & \text{otherwise,} \end{cases}$$

$$(W^{(2)})_{a,b} = \begin{cases} 1 & \text{if } (a, b) \in \{(E, E), (V_0, S_1), (V_1, S_2)\}; \\ -\Omega & \text{if } (a, b) \in \{(C, S_1), (B_{\text{global}}, S_2), (B_{\text{local}}, S_2)\}; \\ \Omega & \text{if } (a, b) = (C, S_2); \\ 0 & \text{otherwise,} \end{cases}$$

$$(W^{(3)})_{a,b} = \begin{cases} 1 & \text{if } (a, b) \in \{(E, E), (S_1, S_1), (S_2, S_1)\}; \\ 0 & \text{otherwise,} \end{cases}$$

$$(W^{(4)})_{a,b} = \begin{cases} 1 & \text{if } (a, b) = (E, S_1); \\ -1 & \text{if } (a, b) = (E, E); \\ 0 & \text{otherwise.,} \end{cases}$$

where $W^{(1)}$ is defined as an identity operator, $W^{(2)}$ is defined to perform the selection, $W^{(3)}$ is defined to sum the terms, $W^{(4)}$ is defined to write the term on scarctchpad to column $E$. $\qquad \square$

## C.2 INCREMENT

**Lemma C.2** (Increment). *If the following conditions hold:*

- *Let the transformer be defined as Definition 3.10.*

- *Let $\widehat{\delta}$ be the nearest representable approximation of the minimum increment angle in position encoding defined in Definition 3.12.*

- *Let $C_1, C_2$ be the index for source position embedding, $D_1, D_2$ be the index for target position embedding.*

*Then we can show that a single-layer transformer can simulate an increment operation, which achieves $X[1, D_1] \leftarrow \sin(\arcsin(X[1, C_1]) + \widehat{\delta})$, $X[1, D_2] \leftarrow \cos(\arccos(X[1, C_2]) + \widehat{\delta})$.*

*Proof.* Because only the MLP layer is needed, we set all the parameters in the attention layer to 0 while keeping the residual connection. Let $S_1, S_2$ be the index for the scratchpad. We construct $W^{(1)}$ corresponding to the rotation matrix, which is defined in Definition 3.12. $W^{(2,3)}$ is defined as identity operator, and $W^{(4)}$ is defined to erase the previous value in $X[:, D]$.

$$(W^{(1)})_{a,b} = \begin{cases} \cos(\widehat{\delta}) & \text{if } (a,b) \in \{(C_1, S_1), (C_2, S_2)\}; \\ -\sin(\widehat{\delta}) & \text{if } (a,b) = (C_1, S_2); \\ \sin(\widehat{\delta}) & \text{if } (a,b) = (C_2, S_1); \\ 1 & \text{if } (a,b) \in \{(D_1, D_1), (D_2, D_2)\}; \\ 0 & \text{otherwise,} \end{cases}$$

$$(W^{(2,3)})_{a,b} = \begin{cases} 1 & \text{if } (a,b) \in \{(D_1, D_1), (D_2, D_2), (S_1, S_1), (S_2, S_2)\}; \\ 0 & \text{otherwise,} \end{cases}$$

$$(W^{(4)})_{a,b} = \begin{cases} 1 & \text{if } (a,b) \in \{(S_1, D_1), (S_2, D_2)\}; \\ -1 & \text{if } (a,b) \in \{(D_1, D_1), (D_2, D_2)\}; \\ 0 & \text{otherwise.} \end{cases}$$

$\square$

### C.3 COMPARISON

**Lemma C.3** (Comparison). *If the following conditions hold:*

- *Let the transformer be defined as Definition 3.10.*

- *Let $\Omega$ represent the maximum absolute value within a clause.*

- *Let $C, D$ be the index for the column for comparing.*

- *Let $E$ be the index for the target column to write the comparison result.*

*Then, we can show that a single-layer transformer can simulate a comparison operation, which achieves writing $X[:, E] \leftarrow X[:, C] < X[:, D]$.*

*Proof.* Because only the MLP layer is needed, we set all the parameters in the attention layer to 0 while keeping the residual connection. Let $S_1, S_2$ be the index for the scratchpad. We construct the weights in MLP as follows:

$$(W^{(1)})_{a,b} = \begin{cases} 1 & \text{if } (a,b) \in \{(E, E), (D, S_1), (D, S_2)\}; \\ -1 & \text{if } (a,b) \in \{(C, S_1), (C, S_2)\}; \\ -\Omega^{-1} & \text{if } (a,b) \in \{(B_{\text{global}}, S_2), (B_{\text{local}}, S_2)\}; \\ 0 & \text{otherwise,} \end{cases}$$

$$(W^{(2)})_{a,b} = \begin{cases} 1 & \text{if } (a,b) = (E, E); \\ \Omega & \text{if } (a,b) = (S_1, S_1); \\ -\Omega & \text{if } (a,b) = (S_2, S_1); \\ 0 & \text{otherwise.} \end{cases}$$

$$(W^{(3)})_{a,b} = \begin{cases} 1 & \text{if } (a,b) \in \{(E, E), (S_1, S_1)\}; \\ 0 & \text{otherwise,} \end{cases}$$

$$(W^{(4)})_{a,b} = \begin{cases} -1 & \text{if } (a,b) = (E, E); \\ 1 & \text{if } (a,b) = (S_1, E); \, , \\ 0 & \text{otherwise.} \end{cases}$$

where $W^{(1)}$ and $W^{(2)}$ are defined to simulate less than function, $W^{(3)}$ is defined as identity layer, $W^{(4)}$ is defined to erase the previous value in $X[:, E]$ and write the term on scarctchpad to column $E$. $\square$

## C.4 READ SCALAR FROM COLUMN

**Lemma C.4** (Read scalar from column). *If the following conditions hold:*

- *Let the transformer be defined as Definition 3.10.*

- *Let $\Omega$ represent the maximum absolute value within a clause.*

- *Let $C_1, C_2$ be the position embedding for source row, $C$ be the row index for the source scalar, and $D$ be the cloumn index for source scalar.*

- *Let $E$ be the index for the target column.*

*Then, we can show that a single-layer transformer can simulate an increment operation, which achieves writing $X[1, E] \leftarrow X[C, D]$.*

*Proof.* The key purpose is to move the information from the local variable to the global variable. The core operation of this construction is to use the hardmax to select the value from a specific position. The position encoding in Definition 3.12 satisfied that $p_i^\top p_j < p_i^\top p_i$ for $i \neq j$.

In the first attention head, we use the standard attention, which is not incorporated the incident matrix to clear the column $E$ preparing for writing:

$$(W_K^{(2)}, W_Q^{(2)})_{a,b} = \begin{cases} 1 & \text{if } (a, b) \in \{(P_1, 1), (P_2, 2), (B_{\text{global}}, 2)\}; \\ 0 & \text{otherwise,} \end{cases}$$

$$(W_V^{(2)})_{a,b} = \begin{cases} -1 & \text{if } (a, b) = (E, E); \\ 0 & \text{otherwise.} \end{cases}$$

where $W_K^{(2)}$ and $W_Q^{(2)}$ are constructed to perform an identity matrix, and $W_V^{(2)}$ is constructed to erase the value in column $E$.

We also need another standard attention head to write the value:

$$(W_K^{(1)}, W_Q^{(1)})_{a,b} = \begin{cases} 1 & \text{if } (a, b) \in \{(P_1, C_1), (P_2, C_2)\}; \\ 0 & \text{otherwise,} \end{cases}$$

$$(W_V^{(1)})_{a,b} = \begin{cases} 2 & \text{if } (a, b) = (D, E) \\ 0 & \text{otherwise.} \end{cases},$$

where $W_K^{(1)}$ and $W_Q^{(1)}$ are constructed to indicate the row, and $W_V^{(1)}$ is constructed to indicate the column.

Noticing that the writing value head writes values in all rows of column $E$. We establish the MLP layer as follows to erase the unnecessary writing:

$$(W^{(1)})_{a,b} = \begin{cases} 1 & \text{if } (a, b) = (E, E); \\ -\Omega & \text{if } (a, b) = (B_{\text{global}}, E); \\ 0 & \text{otherwise,} \end{cases}$$

$$(W^{(2,3)})_{a,b} = \begin{cases} 1 & \text{if } (a, b) = (E, E); \\ 0 & \text{otherwise,} \end{cases}$$

$$(W^{(4)})_{a,b} = \begin{cases} -1 & \text{if } (a, b) = (E, E); \\ 0 & \text{otherwise,} \end{cases}$$

$\square$

## C.5 WRITE SCALAR TO COLUMN

**Lemma C.5** (Write scalar to column). *If the following conditions hold:*

- *Let the transformer be defined as Definition 3.10.*

- *Let $P$ be the index for the position embeddings.*

- *Let $\Omega$ represent the maximum absolute value within a clause.*

- *Let $D$ be the index for source value, i.e., source value is $X[0, D]$.*

- *Let $C_1, C_2$ be the position embedding for target row, $E$ be the index for target column.*

*Then, we can show that a single-layer transformer can simulate the operation of writing a value to a column, i.e., $X[C, E] \leftarrow X[0, D]$.*

*Proof.* The key purpose is to move the information from the global variable to the local variable. The core operation of this construction is to use hardmax to select the value from a specific position. The position encoding in Definition 3.12 satisfied that $p_i^\top p_j < p_i^\top p_i$ for $i \neq j$. Since the MLP layer is not needed, we set all parameters to 0.

First, we construct the first attention layer to write scalar:

$$(W_K^{(1)}, W_Q^{(1)})_{a,b} = \begin{cases} 1 & \text{if } (a,b) \in \{(P_1, C_1), (P_2, C_2)\}; \\ 0 & \text{otherwise,} \end{cases} \qquad (W_V^{(1)})_{a,b} = \begin{cases} 2 & \text{if } (a,b) = (D, E); \\ 0 & \text{otherwise.} \end{cases}$$

where $(W_K^{(1)}$ and $W_Q^{(1)})$ are constructed to find the row $C$, and $W_V^{(1)})$ is constructed to write scalar from column $D$ to column $E$.

The above construction will write some unwanted value to the top row, so we construct another attention head to erase the unwanted value:

$$(W_K^{(2)}, W_Q^{(2)})_{a,b} = \begin{cases} 1 & \text{if } (a,b) \in \{(P_1, 1), (P_2, 2), (B_{\text{global}}, 2)\} \\ 0 & \text{otherwise,} \end{cases}$$

$$(W_V^{(2)})_{a,b} = \begin{cases} -1 & \text{if } (a,b) = (B_{\text{global}}, E); \\ 0 & \text{otherwise.} \end{cases}$$

where we use $B_{\text{global}}$ is used to store global bias. $\qquad\square$

## C.6    TERMINATION

**Lemma C.6** (Termination). *If the following conditions hold:*

- *Let the transformer be defined as Definition 3.10.*

- *Let $P$ be the index for the position embeddings.*

- *Let $\Omega$ represent the maximum absolute value within a clause.*

- *Let $C$ be the index for the executed variable.*

- *Let $E$ be the index for the target column.*

*Then we can show that a single-layer transformer can simulate the operation of writing a value to a column, i.e. $X[1, E] \leftarrow$ (no zero in $X[2 :, C]$).*

*Proof.* We construct the first attention layer to erase the previous value in column $E$:

$$(W_K^{(1)}, W_Q^{(1)})_{a,b} = \begin{cases} 1 & \text{if } (a,b) \in \{(P_1, 1), (P_2, 2), (B_{\text{global}}, 2)\}; \\ 0 & \text{otherwise,} \end{cases}$$

$$(W_V^{(1)})_{a,b} = \begin{cases} -1 & \text{if } (a,b) = (E, E); \\ 1 & \text{if } (a,b) = (B_{\text{global}}, E); \\ 0 & \text{otherwise,} \end{cases}$$

where $W_K^{(1)}$ and $W_Q^{(1)}$ are constructed as identity matrix, $W_V^{(1)}$ is constructed to replace the previous value in column $E$ by 1, which will be used in the following construction.

In the second attention head, we want to construct an attention matrix that can extract information from all the entries of $X[2:, C]$:

$$(W_K^{(2)})_{a,b} = \begin{cases} -1 & \text{if } (a,b) \in \{(B_{\text{global}}, 1), (B_{\text{global}}, 2)\}; \\ 1 & \text{if } (a,b) \in \{(B_{\text{local}}, 1), (B_{\text{local}}, 2)\}; \\ 0 & \text{otherwise}, \end{cases}$$

$$(W_Q^{(2)})_{a,b} = \begin{cases} 1 & \text{if } (a,b) \in \{(B_{\text{global}}, 1), (B_{\text{global}}, 2)\}; \\ -1 & \text{if } (a,b) \in \{(B_{\text{local}}, 1), (B_{\text{local}}, 2)\}; \\ 0 & \text{otherwise}, \end{cases}$$

and the attention matrix is as presented below:

$$\sigma\left(X W_Q^{(2)} W_K^{(2)\top} X^\top\right) = \sigma\left(2 \begin{bmatrix} -1 & 1 & \cdots & 1 \\ \hline 1 & -1 & \cdots & -1 \\ \vdots & \vdots & \ddots & \vdots \\ 1 & -1 & \cdots & -1 \end{bmatrix}\right) = \begin{bmatrix} 0 & \frac{1}{n} & \cdots & \frac{1}{n} \\ \hline 1 & 0 & \cdots & 0 \\ \vdots & \vdots & \ddots & \vdots \\ 1 & 0 & \cdots & 0 \end{bmatrix}.$$

We construct the value matrix as:

$$(W_V^{(2)})_{a,b} = \begin{cases} \Omega & \text{if } (a,b) = (C, E); \\ -\Omega & \text{if } (a,b) = (B_{\text{local}}, E); \\ 0 & \text{otherwise}, \end{cases}$$

after this, the top entry of column $E$ is $1 - \Omega + \frac{\Omega}{n} \sum_i (X[i, C])$. Applying ReLU units, we construct the MLP layer as follows:

$$(W^{(1)})_{a,b} = \begin{cases} 1 & \text{if } (a,b) \in \{(E, E), (E, S_1)\} \\ -1 & \text{if } (a,b) = (E, S_2); \\ 0 & \text{otherwise}, \end{cases}$$

$$(W^{(2,3)})_{a,b} = \begin{cases} 1 & \text{if } (a,b) \in \{(E, E), (S_1, S_1), (S_2, S_2)\}; \\ 0 & \text{otherwise}, \end{cases}$$

$$(W^{(4)})_{a,b} = \begin{cases} 1 & \text{if } (a,b) \in \{(E, E), (S_2, E)\}; \\ -1 & \text{if } (a,b) = (S_1, E); \\ 0 & \text{otherwise}. \end{cases}$$

which writes $\phi(1 - \Omega + \frac{\Omega}{n} \sum_i (X[i, C]))$ to $X[1, E]$. It's easy to know that if only if all the value are 1 in $X[2:, C]$, $X[1, E]$ gets value 1. $\qquad \square$

## C.7 READ FROM INCIDENT MATRIX

**Lemma C.7** (Read from incident matrix). *If the following conditions hold:*

- *Let the transformer be defined as Definition 3.10.*

- *Let $P$ be the index for the position embeddings.*

- *Let $C$ be the index for the source row/column index.*

- *Let $D$ be the index for the target column.*

- *Let $P_{\text{cur}}$ be the index for the position embedding of row/column $C$.*

*Then we can show that:*

- **Part 1.** *A single-layer transformer can simulate the operation of reading a row from A, i.e. $X[:, D] \leftarrow \widetilde{A}[C, :]$.*

- **Part 2.** *A single-layer transformer can simulate the operation of reading a column from A, i.e. $X[:, D] \leftarrow \widetilde{A}[:, C]$.*

*Proof.* Following from Definition 3.6, we can either employ $\psi^{(i)}(X, \widetilde{A})$ or $\psi^{(i)}(X, \widetilde{A}^\top)$. So, reading a column and reading a row is equivalent in our setting. Considering the reading row case, we construct one attention head as follows:

$$(W_K^{(1)}, W_Q^{(1)})_{a,b} = \begin{cases} 1 & \text{if } (a,b) \in \{(P_1, (P_{\text{cur}})_1), (P_2, (P_{\text{cur}})_2)\}; \\ 0 & \text{otherwise,} \end{cases}$$

$$(W_V^{(1)})_{a,b} = \begin{cases} 2 & \text{if } (a,b) = (B_{\text{global}}, D); \\ 0 & \text{otherwise,} \end{cases}$$

where $W_Q^{(1)}$ and $W_K^{(1)}$ are constricuted to get row $C$ following from Definition 3.12. $W_V^{(1)}$ is used to move row $C$ of matrix $A$ to column $D$ of matrix $X$.

For the second attention head, we use a standard attention head to erase the previous value in column $D$.

$$(W_K^{(2)}, W_Q^{(2)})_{a,b} = \begin{cases} 1 & \text{if } (a,b) \in \{(P_1, 1), (P_2, 2), (B_{\text{global}}, 2)\}; \\ 0 & \text{otherwise,} \end{cases}$$

$$(W_V^{(2)})_{a,b} = \begin{cases} -1 & \text{if } (a,b) = (D, D); \\ 0 & \text{otherwise,} \end{cases}$$

Where $W_K^{(2)}$, $W_Q^{(2)}$ are constructed to make the attention matrix as identity matrix, $W_V^{(2)}$ is constructed to erase value. For the MLP layer, we just make it as the residential connection by setting all parameters to 0. $\square$

## C.8 AND

**Lemma C.8** (AND). *If the following conditions hold:*

- *Let the transformer be defined as Definition 3.10.*

- *Let $\Omega$ represent the maximum absolute value within a clause.*

- *Let $C$ and $D$ be the index for the executed variable.*

- *Let $E$ be the index for the target column.*

*Then we can show that a single-layer transformer can simulate the operation of AND, i.e.* $X[1, E] \leftarrow X[1, C] \land X[1, D]$.

*Proof.* In this construction, we want to have $\phi(X[1, C] + X[1, D] - 1)$, so that if and only if $X[1, C] = 1$ and $X[1, D] = 1$, we have $\phi(X[1, C] + X[1, D] - 1) = 1$, otherwise $\phi(X[1, C] + X[1, D] - 1) = 0$. Because only the MLP layer is needed, we set all the parameters in the attention layer to 0 while keeping the residual connection. We construct MLP layers as follows:

$$(W^{(1)})_{a,b} = \begin{cases} 1 & \text{if } (a,b) \in \{(E, E), (C, S), (D, S)\}; \\ -1 & \text{if } (a,b) = (B_{\text{global}}, S); \\ 0 & \text{otherwise,} \end{cases}$$

$$(W^{(2,3)})_{a,b} = \begin{cases} 1 & \text{if } (a,b) \in \{(E, E), (S, S)\}; \\ 0 & \text{otherwise,} \end{cases}$$

$$(W^{(4)})_{a,b} = \begin{cases} -1 & \text{if } (a,b) = (E, E); \\ 1 & \text{if } (a,b) = (S, E); \\ 0 & \text{otherwise,} \end{cases}$$

where $W^{(1)}$ is the core of the construction, $W^{(2,3)}$ are defined as an identity operator, $W^{(4)}$ is used to erase the previous value and move the result in scratchpad to column $E$.

$\square$

## C.9 REPEAT AND

**Lemma C.9** (Repeat AND). *If the following conditions hold:*

- *Let the transformer be defined as Definition 3.10.*

- *Let $\Omega$ represent the maximum absolute value within a clause.*

- *Let $C$ be the index for a Boolean value.*

- *Let $D$ and $E$ be the index for two columns, where each entry is a Boolean value.*

*Then we can show that a single-layer transformer can simulate the operation of repeat AND, i.e.,*
$X[i, D] \leftarrow X[1, C] \wedge X[i, D] \wedge \neg X[i, E]$ *for $i \in \{2, 3, \cdots, K\}$.*

*Proof.* In this construction, we only use MLP layers. For multi-head attention layers, we just make it as the residential connection by setting all parameters to $0$.

$$
(W^{(1)})_{a,b} = \begin{cases} 1 & \text{if } (a,b) \in \{(D, D), (C, D), (D, S_1)\}; \\ -1 & \text{if } (a,b) \in \{(E, D), (B_{\text{global}}, D)(B_{\text{local}}, D)\}; \\ 0 & \text{otherwise}, \end{cases}
$$

$$
(W^{(2,3)})_{a,b} = \begin{cases} 1 & \text{if } (a,b) \in \{(D, D), (S_1, S_1)\}; \\ 0 & \text{otherwise}, \end{cases}
$$

$$
(W^{(4)})_{a,b} = \begin{cases} 1 & \text{if } (a,b) = (D, D); \\ -1 & \text{if } (a,b) = (S_1, D); \\ 0 & \text{otherwise}. \end{cases}
$$

where $(W^{(1)})$ is used to construct $\phi(X[1, C] + X[i, D] - X[i, E] - 1)$, $(W^{(2,3)})$ are constructed as identity layers, $(W^{(4)})$ is constructed to erase previous value in column $D$. $\qquad \square$

## C.10 REPEAT ADDITION

**Lemma C.10** (Repeat addition). *If the following conditions hold:*

- *Let the transformer be defined as Definition 3.10.*

- *Let $\Omega$ represent the maximum absolute value within a clause.*

- *Let $P$ be the index for the position embeddings.*

- *Let $C$ be the index for a scalar.*

- *Let $D$ be the index for a column.*

*Then we can show that a single-layer transformer can simulate the operation of repeat addition, i.e.*
$X[:, D] \leftarrow \mathbf{1}_K \cdot X[1, C] + X[:, D]$.

*Proof.* We build the first attention head as:

$$
(W_K^{(1)})_{a,b} = \begin{cases} 1 & \text{if } (a,b) \in \{(B_{\text{global}}, 1), (B_{\text{global}}, 2)\}; \\ 0 & \text{otherwise}, \end{cases}
$$

$$
(W_Q^{(1)})_{a,b} = \begin{cases} 1 & \text{if } (a,b) \in \{(B_{\text{global}}, 1), (B_{\text{global}}, 2), (B_{\text{local}}, 1), (B_{\text{local}}, 2)\}; \\ 0 & \text{otherwise}, \end{cases}
$$

$$
(W_V^{(1)})_{a,b} = \begin{cases} 1 & \text{if } (a,b) = (C, D); \\ 0 & \text{otherwise}, \end{cases}
$$

where we have

$$
\sigma\left(XW_Q^{(1)}W_K^{(1)\top}X^\top\right) = \sigma\left(2\begin{bmatrix} 1 & 0 & \cdots & 0 \\ 1 & 0 & \cdots & 0 \\ \vdots & \vdots & \ddots & \vdots \\ 1 & 0 & \cdots & 0 \end{bmatrix}\right) = \begin{bmatrix} 1 & 0 & \cdots & 0 \\ 1 & 0 & \cdots & 0 \\ \vdots & \vdots & \ddots & \vdots \\ 1 & 0 & \cdots & 0 \end{bmatrix}.
$$

Using the first attention head, we can repeat $X[1, C]$ as a column. For the second attention head, we just select the column to erase the first value in column $D$.

$$
(W_K^{(2)}, W_Q^{(2)})_{a,b} = \begin{cases} 1 & \text{if } (a,b) \in \{(P_1, 1), (P_2, 2), (B_{\text{global}}, 2)\}; \\ 0 & \text{otherwise,} \end{cases}
$$

$$
(W_V^{(2)})_{a,b} = \begin{cases} -1 & \text{if } (a,b) = (C, D); \\ 0 & \text{otherwise.} \end{cases}
$$

Where $W_K^{(2)}$, $W_Q^{(2)}$ are constructed to make the attention matrix as an identity matrix, $W_V^{(2)}$ is constructed to erase value. For the MLP layer, we just make it as the residential connection by setting all parameters to 0. $\qquad\square$

---

**Algorithm 3** Iteration of minimum value

---

1: $\text{idx}_{\text{cur}}$, $\text{val}_{\text{cur}} \leftarrow 0, 0$
2: $\text{idx}_{\text{best}}$, $\text{val}_{\text{best}} \leftarrow 0, \Omega$
3: $\text{visit}_{\text{min}} \leftarrow \mathbf{0}_{n_v} \| \mathbf{1}_{K-1-n_v}$
4: $\text{termination}_{\text{min}} \leftarrow 0$           ▷ Initialization of minimum value
5: ————————————————————————————————————
6: **procedure** GETMINIMUMVALUE($x \in \mathbb{R}^d$)
7:     **if** $\text{termination}_{\text{min}} = 1$ **then**
8:         $\cdots$         ▷ Re-initialization of minimum value, Lemma C.1
9:         $\text{termination}_{\text{min}} \leftarrow 0$
10:     **end if**
11:     $\text{idx}_{\text{cur}} \leftarrow \text{idx}_{\text{cur}} + 1$         ▷ Increment, Lemma C.2
12:     $\text{val}_{\text{cur}} \leftarrow x[\text{idx}_{\text{cur}}]$        ▷ Read scalar from column, Lemma C.4
13:     **if** $\text{val}_{\text{cur}} < \text{val}_{\text{best}}$ **then**        ▷ Compare, Lemma C.3
14:         $\text{val}_{\text{best}}$, $\text{idx}_{\text{best}} \leftarrow \text{val}_{\text{cur}}$, $\text{idx}_{\text{cur}}$     ▷ Update variables, Lemma C.1
15:     **end if**
16:     $\text{visit}_{\text{min}}[\text{idx}_{\text{cur}}] \leftarrow 1$       ▷ Write scalar to column, Lemma C.5
17:     $\text{termination}_{\text{min}} \leftarrow \neg(0 \text{ in visit}_{\text{min}})$     ▷ Trigger termination, Lemma C.6
18: **end procedure**

---

## D   MISSING PROOF IN SIMULATION

**Theorem D.1** (Dijkstra's Algorithm on hypergraph, formal version of Theorem 6.2). *A looped transformer $h_T$ exists, where each layer is defined as in Definition 3.10, consisting of 27 layers, where each layer includes 3 attention heads with feature dimension of $O(1)$. This transformer simulates Dijkstra's Algorithm iteratively for weighted hypergraphs, supporting up to $O(\widehat{\delta}^{-1})$ vertices and $O(\widehat{\delta}^{-1})$ hyperedges.*

*Proof.* Let Dijkstra's algorithm be considered as described in Algorithm 4. Following from Lemma 6.1 and Lemma B.1, the operation of iteratively visiting vertices and hyperedges requires 18 layers, as established in these lemmas. For the remaining part of the algorithm, the operations include 4 selection operations, 1 add operation, 1 compare operation, 1 repeat AND operation, 1 write scalar to column operation, and 1 trigger termination operation. According to Lemma C.1, Lemma C.10, Lemma C.9, Lemma C.3, Lemma C.5, and Lemma C.6, these operations together require 9 layers. Therefore, the total number of layers required to simulate Dijkstra's algorithm is:

$$18 + 9 = 27$$

layers of the transformer. $\qquad\square$

---

**Algorithm 4** Dijkstra's algorithm for shortest path

---

1: **procedure** HYPERGRAPHDIJKSTRA($A \in \mathbb{R}_+^{n_v \times n_e}$, start $\in \mathbb{N}$)
2:     prev, dists, dists$_{\text{masked}}$, changes, iszero $\leftarrow \mathbf{0}_{K-1}$
3:     visit, dists, prev $\leftarrow \mathbf{0}_{n_v} \| \mathbf{1}_{K-1-n_v}, \Omega \cdot \mathbf{1}_{K-1}, [K-1]$
4:     visit[start], termination $\leftarrow 0$                     ▷ Initialization of minimum value and visiting hyperedges

---

6:     **while** termination is **false do**
7:         **for** $i = 1 \rightarrow K-1$ **do**
8:             **if** visit[$i$] is **true then**
9:                 dists$_{\text{masked}}$[$i$] $\leftarrow \Omega$                     ▷ Selection, Lemma C.1
10:            **else**
11:                dists$_{\text{masked}}$[$i$] $\leftarrow$ dists[$i$]
12:            **end if**
13:        **end for**

---

15:        GETMINIMUMVALUE (dists$_{\text{masked}}$)                     ▷ Get minimum value, Lemma B.1

---

17:        **if** termination$_{\text{min}}$ is **true then**
18:            node $\leftarrow$ idx$_{\text{best}}$
19:            dist $\leftarrow$ val$_{\text{best}}$                     ▷ Selection, Lemma C.1
20:        **end if**

---

22:        VISITHYPERDEGE ($A$)                     ▷ Degradation, Theorem 4.1

---

24:        **if** termination$_{\text{hyperedge}}$ = 1 **then**
25:            candidates $\leftarrow$ candidates$_{\text{hyperedge}}$                     ▷ Selection, Lemma C.1
26:        **end if**
27:        **for** $i = 1 \rightarrow K-1$ **do**
28:            candidates[$i$] $\leftarrow$ candidates[$i$] + dist                     ▷ Addition, Lemma C.10
29:        **end for**
30:        **for** $i = 1 \rightarrow K-1$ **do**
31:            changes[$i$] $\leftarrow$ candidates[$i$] < dists[$i$]                     ▷ Compare, Lemma C.3
32:        **end for**
33:        **for** $i = 1 \rightarrow K-1$ **do**
34:            **if** termination$_{\text{min}}$ = 0 **and** iszero[$i$] = 1 **then**
35:                changes[$i$] $\leftarrow 0$                     ▷ repeat AND, Lemma C.9
36:            **end if**
37:        **end for**
38:        **for** $i = 1 \rightarrow K-1$ **do**
39:            **if** changes[$i$] = 1 **then**
40:                prev[$i$], dists[$i$] $\leftarrow$ node, candidates[$i$]                     ▷ Selection. Lemma C.1
41:            **end if**
42:        **end for**
43:        visit[node] $\leftarrow$ visit[node] + termination$_{\text{min}}$                     ▷ Write scalar to column, Lemma C.5
44:        termination $\leftarrow \neg(0$ in visit$)$                     ▷ Trigger termination, Lemma C.6
45:    **end while**
46:    **return** prev, dists
47: **end procedure**

---

**Lemma D.2** (Visiting hyperedges iteratively, formal version of Lemma 6.1). *A looped transformer $h_T$ exists, where each layer is defined as in Definition 3.10, consisting of 10 layers where each layer includes 3 attention heads with feature dimension of $O(1)$. This transformer simulates the operation of iteratively visiting hyperedges for weighted hypergraphs, accommodating up to $O(\widehat{\delta}^{-1})$ vertices and $O(\widehat{\delta}^{-1})$ hyperedges.*

*Proof.* Let the operation of visiting hyperedges iteratively be defined as described in Algorithm 2. This operation requires 1 increment operation, which requires single layer transformer to construct the following from Lemma C.2, 2 compare operations, which require 3 layers transformer to construct the following from Lemma C.3, 3 selection operations, which require 3 layers transformer to construct following from Lemma C.1, 1 AND operation which require single layer transformer to construct following from Lemma C.8, 1 read-from-incident-matrix operation which requires single

---

**Algorithm 5** Helly, Algorithm 3 in (Bretto, 2013)

---

1: $\text{idx}_x \leftarrow 0$
2: $\text{idx}_y \leftarrow 1$
3: $\text{idx}_v \leftarrow 0$
4: $\text{termination} \leftarrow 0$
5: $\text{helly} \leftarrow 1$             ▷ Initialization of Helly
6: ————————————————————————————————
7: **while** $\text{termination} = 0$ **do**
8:      $\text{hyperedge}_x \leftarrow A[:, \text{idx}_x]$
9:      $\text{hyperedge}_y \leftarrow A[:, \text{idx}_y]$
10:      $\text{hyperedge}_v \leftarrow A[:, \text{idx}_v]$        ▷ Read column from incident matrix, Lemma C.7
11:      $\text{hyperedge}_x \leftarrow \text{hyperedge}_x > 0$
12:      $\text{hyperedge}_y \leftarrow \text{hyperedge}_y > 0$
13:      $\text{hyperedge}_v \leftarrow \text{hyperedge}_v > 0$        ▷ Compare, Lemma C.3
14:      $\text{intersection} \leftarrow \text{hyperedge}_x \wedge \text{hyperedge}_y \wedge \text{hyperedge}_v$       ▷ AND, Lemma C.8
15:      $\text{helly}_v \leftarrow \neg(1 \text{ in intersection})$
16:      **if** $\text{helly}_v = 1$ **then**
17:          $\text{helly} \leftarrow 0$
18:          $\text{termination} \leftarrow 1$        ▷ Selection, Lemma C.1
19:      **end if**
20:      $\text{idx}_v \leftarrow \text{idx}_v + 1$        ▷ Increment, Lemma C.2
21:      **if** $\text{idx}_v > n_v$ **then**
22:          $\text{idx}_v \leftarrow 1$        ▷ Selection, Lemma C.1
23:          $\text{idx}_y \leftarrow \text{idx}_y + 1$        ▷ Selection and Increment, Lemma C.1 and Lemma C.2
24:      **end if**
25:      **if** $\text{idx}_y > n_v$ **then**
26:          $\text{idx}_x \leftarrow \text{idx}_x + 1$
27:          $\text{idx}_y \leftarrow \text{idx}_x + 1$        ▷ Selection and Increment, Lemma C.1 and Lemma C.2
28:      **end if**
29:      **if** $\text{idx}_x = n_v$ **then**
30:          $\text{termination} \leftarrow 1$        ▷ Selection, Lemma C.1
31:      **end if**
32: **end while**
33: **return** helly

---

layer transformer to construct following from Lemma C.7, 1 write-scalar-to-column operation which requires single layer transformer to construct following from Lemma C.5, and 1 trigger-termination operation which require single layer transformer to construct following from Lemma C.6. The whole algorithm can be constructed by

$$1 + 1 + 1 + 1 + 1 + 2 + 3 = 10$$

layers of the transformer. □

# E    MISSING PROOF IN MAIN RESULTS

We are now ready to show our main results based on our previous components.

## E.1    DEGRADATION

**Theorem E.1** (Degradation, formal version of Theorem 4.1). *A looped transformer $h_T$ defined in Definition 3.11 exists, consisting of 10 layers, where each layer includes 3 attention heads with feature dimension of $O(1)$. This transformer can simulate the degradation operation (Algorithm 2) for hypergraphs, supporting up to $O(\widehat{\delta}^{-1})$ vertices and $O(\widehat{\delta}^{-1})$ hyperedges, where $\widehat{\delta}$ defined in Definition 3.12 is the nearest representable approximation of the minimum increment angle.*

*Proof.* Using the same construction as Lemma 6.1, we can show that 10 layers can iterate the algorithm. Furthermore, for an incident matrix, we require its dimensions to be within the smallest computable precision. Therefore, it is necessary to ensure that there are at most $O(\widehat{\delta}^{-1})$ vertices and $O(\widehat{\delta}^{-1})$ hyperedges. □

### E.2 HELLY

**Theorem E.2** (Helly, formal version of Theorem 4.2)**.** *A looped transformer $h_T$ exists, where each layer is defined as in Definition 3.10, consisting of 11 layers, where each layer includes 3 attention heads with feature dimension of $O(1)$. This transformer can simulate the Helly algorithm (Algorithm 5) for weighted hypergraphs, handling up to $O(\widehat{\delta}^{-1})$ vertices and $O(\widehat{\delta}^{-1})$ hyperedges, where $\widehat{\delta}$ defined in Definition 3.12 is the nearest representable approximation of the minimum increment angle.*

*Proof.* This algorithm requires 3 increment operations, 5 selection operations, 1 AND operation, 1 compare operation, and 1 read-from-incident-matrix operation. By applying Lemma C.2, Lemma C.1, Lemma C.8, Lemma C.3, and Lemma C.7, each of these operations can be constructed using a single-layer transformer, as detailed in the corresponding lemmas. The total number of layers required for the entire degradation operation is: $3 + 5 + 1 + 1 + 1 = 11$. Thus, the degradation operation can be constructed using 11 layers of transformer. Furthermore, for an incident matrix, we require its dimensions to be within the smallest computable precision. Therefore, it is necessary to ensure that there are at most $O(\widehat{\delta}^{-1})$ vertices and $O(\widehat{\delta}^{-1})$ hyperedges. □

## F IMPACT STATEMENTS

This research shows how Looped Transformers can learn to perform complex calculations on hypergraphs, which map intricate relationships between many items. This could lead to more powerful AI tools for solving complex problems. As this work is theoretical and focuses on the capability of these models, we don't foresee direct negative societal impacts.

## LLM USAGE DISCLOSURE

LLMs were used only to polish language, such as grammar and wording. These models did not contribute to idea creation or writing, and the authors take full responsibility for this paper's content.

