# OpenReview forum: "Neural Algorithmic Reasoning for Hypergraphs with Looped Transformers"
_ICLR.cc/2026/Conference — Submitted to ICLR 2026_

### Official Review · Reviewer_cQi6 · 2025-10-29

**Soundness:** 3
**Presentation:** 1
**Contribution:** 2
**Rating:** 2
**Confidence:** 4

**Summary:**

The paper demonstrates that looped transformers (which iteratively stack transformers with shared parameters) can represent certain hypergraph reasoning tasks (such as Dijkstra's shortest path and Helly's algorithm) using a constant-sized transformer sub-unit. A provided hypergraph is incorporated into the transformer as an padded incident matrix that multiplies the output of a hardmax attention unit. The bit-precision must scale logarithmically with the size of the hypergraph.

Theorem 4.1 is a _degradation theorem_, which shows that a constant-size transformer can access adjacency relationships between pairs of nodes with the provided incidence matrix. By combining this theorem with prior results, hypergraph generalizations of common graph problems can be solved with looped transformers.

Theorem 4.2 shows that the Helly property can be detected in a hypergraph in a similar scaling regime.

**Strengths:**

The paper covers a novel setting and answers natural theoretical questions in that setting. There are several interesting ideas in the paper that are worthy of investigation:
* The representation of a hypergraph's incidence matrix in the attention unit;
* The incorporation of the combinatorial Helly problem alongside graph benchmarks;
* The relationship uncovered between the incidence matrix and efficient adjacency look-ups;
* The detailed enumeration of reasoning primitives and their assembly into discrete algorithms.

As far as I am aware, the theorems are sound and the transformer is precisely defined.

**Weaknesses:**

The results and their surrounding narrative provide little contextualization or motivation for the theoretical or empirically-minded reader.

Without empirical results, it is difficult to understand whether the constructions and logical circuitry described herein are plausibly learnable. The scaling of the Looped Transformer architecture---where the width is bounded by a constant, but the recurrent unit repeats a polynomial number of times in the size of the hypergraph---is unrealistic. This extreme depth scaling, combined with the hardmax activation and discrete circuit simulation, indicate that training a model in this setting would likely be impossible.

If the aim is to present the results as a purely theoretical paper, than other conceptual anchors are essential for understanding the significance of the constructions. For instance, it would be valuable to understand whether these tasks can be solved by other architectures, whether they can be contrasted with other reasoning tasks, or whether generalization results are possible. Without such contextualization, it is difficult to understand the significance of the results or how they fit into a broader literature.

Several points in the paper body and appendix indicate very rushed writing with frequent typo and unclear explanations. For instance:
* The degradation algorithm is presented as a large block of pseudocode with limited explanation in the paper body. The pseudocode is not clear about what is accomplished in each loop of the algorithm; it is not apparent what "hyperedge" and "min" are defined in the block.
* The operator explanations in Section 5 are described vaguely, when clearer mathematical definitions are possible. It's unclear how different components (e.g. the arrays in "Selection") are encoded in the weights, hypergraph, or input vectors; they do not help with an understanding of the degradation algorithm.
* The provided implementation of Helly's algorithm is a complex indexing and logic sequence that is not clearly explained or motivated.
* The related work section at the start of the appendix presents a long list of papers without narrative.

I think future revisions of this paper could be compelling, given a more complete contextualization of the problems and architectures, empirical investigations, and more comprehensive editing.

Minor notes / typos:
- Is there a reason that the positional encoding needs to be defined recursively? Is it equivalent to say $p_i = (\sin(i \hat\delta), \cos(i\hat\delta))$?
- Spelling: "Furthere" [359], "Comparason" [406].
- I would recommend citing the following paper, which discusses the powers of fixed-depth looped transformers. https://arxiv.org/abs/2503.03961

**Questions:**

Are there other hypergraph problems that cannot be solved in this constant-size looped transformer setting?

How do these results compare with other representational results for graph tasks on transformers, GNNs, and other models?

Does the appearance of the incidence matrix in the attention unit make the model more akin to a message-passing GNN-transformer hybrid than a transformer itself? If so, how can this be benchmarked among notions of graph and hypergraph message-passing complexity (e.g. WL tests)?

---

### Official Review · Reviewer_d2fr · 2025-10-31

**Soundness:** 3
**Presentation:** 3
**Contribution:** 3
**Rating:** 6
**Confidence:** 2

**Summary:**

This paper extends the theoretical framework of looped transformers, previously shown to simulate classical graph algorithms—to the more general setting of hypergraphs. The authors introduce a dynamic degradation mechanism to map hypergraph incidence structures to adjacency-like forms, enabling the simulation of standard algorithms such as Dijkstra, BFS, and DFS. They also propose a hyperedge-aware encoding scheme that allows the simulation of hypergraph-specific algorithms, such as Helly’s algorithm. Theoretical results guarantee that these simulations can be performed with constant-width transformers, suggesting that looped transformers may serve as general-purpose neural algorithmic solvers for higher-order relational structures.

**Strengths:**

1) The paper is well written and easy to follow despite the heavy theoretical content.

2) The mathematical formalization is precise and consistent.

3) The logical flow is clear: the reader can easily understand the motivation and the main theoretical results.

**Weaknesses:**

**Novelty** The paper represents a structured extension of an existing theoretical framework [1]. While the move from graphs to hypergraphs is non-trivial and technically well executed, the conceptual novelty is limited. Can you pleas clarify more explicitly how this work positions itself with respect to that prior paper?

**Experimental validation** I understand that this paper is heavily theoretical, and experiments may not be strictly required. However, I have a question rather than a criticism: are there any experimental or empirical ways to verify the theoretical part? For example, could one instantiate your looped transformer and empirically observe the step-by-step simulation of the algorithms you formalize?

**Practical relevance** Related to the above, it would help to better explain in which kinds of tasks the hypergraph algorithms simulated by your looped transformer could be useful. For instance, are there specific domains (e.g., combinatorial optimization, higher-order relational reasoning, molecular modeling, etc.) where this theoretical capacity would translate into practical benefits?

[1] de Luca, Artur Back, and Kimon Fountoulakis. "Simulation of Graph Algorithms with Looped Transformers." Forty-first International Conference on Machine Learning.

**Questions:**

See weaknesses.

---

### Official Review · Reviewer_Y5dx · 2025-10-31

**Soundness:** 3
**Presentation:** 2
**Contribution:** 2
**Rating:** 2
**Confidence:** 5

**Summary:**

This paper studies the expressivity of looped Transformers for algorithms on hypergraphs. The main contribution is a set of constructive results showing that such models can simulate classical graph/hypergraph algorithms, including Dijkstra’s shortest paths, BFS, and DFS.

**Strengths:**

The theoretical development appears careful and rigorous, with clear statements of theorems and proofs.

Establishing expressivity for hypergraph algorithms is a natural and timely question, and the results could be of interest to the algorithmic reasoning community.

**Weaknesses:**

The paper is entirely theoretical. While I appreciate the focus, even a small empirical demonstration would strengthen the case that the abstractions matter in practice (e.g., whether a looped Transformer can learn to execute hypergraph Dijkstra on synthetic tasks; sample efficiency; robustness to noise; comparison to non-looped baselines). In the current form, the bar for acceptance rests solely on theoretical novelty and significance.

Many variations of BFS/DFS/Dijkstra are already known to be within reach of highly expressive sequence models. It would help to sharpen what is surprising here: Is looping essential? Does the hypergraph setting reveal capabilities that standard Transformers provably lack, or significantly reduce resource requirements (depth/width/sequence length)? Clearer comparisons to prior expressivity results would help assess the incremental contribution.

The paper would benefit from a careful proofreading pass; there are a number of typos and minor grammatical issues that detract from readability.

**Questions:**

Do the authors expect their constructions to be learnable?  Is there any theoretical evidence that looped transformers can provably learn hypergraph algorithms when trained with sgd in polynomial time?

---

### Official Review · Reviewer_qRDA · 2025-10-31

**Soundness:** 3
**Presentation:** 2
**Contribution:** 3
**Rating:** 4
**Confidence:** 2

**Summary:**

The paper studies looped transformers (here, essentially a standard transformer layer with an additional attention mechanism that incorporates the incident matrix applied in a recurrent way) for solving algorithms over hypergraphs. In particular, one hypergraph-specific algorithm is investigated. Moreover the authors describe how to consider standard graph algorithms (e.g., BFS) in this setting.

**Strengths:**

- Neural algorithmic reasoning and hypergraphs are both interesting and important topics
- The paper proposes valid next steps in this domain:
It extends the results from Back de Luca et al. to port the graph algorithms to hypergraphs (which is simple). Beyond that, it presents one hyper graph-related algorithm as an example and describes the power of simulating deterministic hypergraph motif algorithms more generally. This is a decent contribution overall.
- It's a good idea to cover the simulation of smaller key operations separately. This makes the findings potentially more generally applicable, in algorithms beyond the ones discussed.

**Weaknesses:**

- The paper is hard to read for readers not familiar with the domain. For instance, the paper jumps from the transformer definition in the preliminaries to Algorithm 2, without specifying the relationship (beyond the definition of a simulation). The introduction, related work, or preliminaries would be good places for giving a basic introduction.
- More generally, technical details/definitions are often given without any intuition. Especially in a graph/algorithmic setting, pictures could provide useful visualizations.

I'm giving the rating for these reasons, because they make me unable to fully judge the validity of the results in reasonable time and without much external reading.


Minor:
- 4.3 "further"
- L. 406 Comparison.
- "degradation algorithm" sounds strange to me - why degradation?

**Questions:**

----------------------------------

---

### Meta-Review · Area_Chair_8CCm · 2026-01-04

**Summary:**

Several crucial concerns have been raised by the reviewers, including low presentation quality, lack of empirical evaluation, and the unclear significance of the theoretical results. I have carefully read the paper and agree with the reviewers on these points. As there was no rebuttal from the authors, these concerns remain unaddressed. I therefore recommend rejection of the paper.

**Reviewer Concerns:**

Since there was no rebuttal from the authors, the reviewers’ concerns remain outstanding.

**Reviewer Scores:**

Since there was no rebuttal from the authors, active discussion between the authors and reviewers is unlikely to occur. Therefore, the reviewers are unlikely to change their scores.

---

### Decision · Program_Chairs · 2026-01-26

Reject